# Climate Insurance for Agriculture in Europe: On the Merits of Smart Contracts and Distributed Ledger Technologies

**Reimund Schwarze *** and **Oleksandr Sushchenko**

Department of Economics, Helmholtz-Centre for Environmental Research (UFZ), Permoser Strasse 15, 04318 Leipzig, Germany; sushchenko@europa-uni.de
* Correspondence: reimund.schwarze@ufz.de

**Abstract:** Climate insurance has become a crucial issue due to the increasing number of climate-related catastrophic events and the associated losses for the economy in general and insurance companies in particular. The extremely hot and dry summers of 2018 and 2019 in some European countries highlighted existing weaknesses in European agricultural insurance mechanisms, with farmers having to wait for months before compensation payments could be made. Our paper compares features of yield-based insurance and index-based insurance (IBI) in agriculture in the light of new developments and trends in information technology (IT). The results show that applying Distributed Ledger Technologies (DLT) in combination with IBI could not only resolve existing problems but also facilitate the development of innovative risk management tools under the EU's Common Agricultural Policy (CAP) post-2020 reform.

**Keywords:** distributed ledger technology; index-based insurance; climate insurance; smart contracts; e-agriculture; CAP post-2020 reform

**JEL Classification:** G22; L8; O52

## 1. Introduction

Climate change is associated with a number of adverse phenomena, including extreme weather events, natural disasters and others of a similar kind. It poses risks to economic development and requires additional expenditure to prevent catastrophic events or to compensate for damages already caused. The World Economic Forum's (WEF) Global Risk Report 2022 recognizes climate change and its disruptive consequences as being the greatest risk to economic activity (WEF 2022). Due to the increasing number of climate-related extreme weather events, natural disasters and the associated losses, climate insurance has already garnered considerable attention and become an important issue for the economy in general and for the insurance industry in particular. The extremely hot summers of 2018 and 2019 in Europe demonstrated that existing approaches to agricultural insurance have numerous shortcomings (e.g., farmers had to wait for months to have their claims settled, not in the least due to the overly bureaucratic processes involved). For insurance companies, climate change poses new challenges while at the same time opening up new opportunities for the development of innovative financial products. In addition, institutional investors (e.g., pension and mutual funds) offer innovative instruments (e.g., catastrophe bonds) that provide opportunities to transfer climate-related risks to the financial markets (Hagendorff et al. 2014; Morana and Sbrana 2019). From the point of view of the agricultural insurance industry, there is a large number of so-called index-based insurance solutions (IBI)[1] that are alternatives to yield-based insurance[2]. The main advantage of IBI is the use of an independent and objective physical indicator which makes it possible to overcome existing problems in traditional crop loss insurance and to achieve potential cost savings (Jarrod et al. 2018). Nevertheless, some technical problems of applying IBI in agriculture (e.g., data

collection and processing) and the issue of high cost remain unresolved. These bottlenecks could partly be eased by implementing Distributed Ledger Technologies (DLT). The now widespread use of DLT in the crypto-currency market has highlighted some positive features of this IT solution and opened up possible options for it to be applied as a technical facilitator in the financial market (e.g., fintech and insurtech services[3]). DLT may turn out to be one of the key technical solutions to assist in connecting technologies on the corporate level with such innovations as wearables, drones and devices connected to the Internet of Things. This IT solution could potentially accelerate transformations across insurance services and capital distribution (KPMG 2017). In addition, DLT-based platforms could improve resilience to disasters and speed up recovery efforts after them by enabling decentralized storage of the critical information required to file the claims (FEMA 2019).

In this regard, a set of research questions arise. Our first research question is whether or not IBI is a better solution for agricultural risks than yield-based insurance. Second, could the use of DLT result in substantial time and cost savings for insurance services? Additionally, if this is the case, could an IBI-based climate insurance scheme for agriculture on the EU level (that makes use of DLT) improve the European Union's existing Agricultural Policy?

## 2. State of the Literature

In the last decade, research on agricultural IBI has advanced rapidly. Most of the applied work has focused primarily on the barriers for scaling up and the diffusion of IBI in developing countries (Hazell et al. 2017; Binswanger-Mkhize 2012; Greatrex et al. 2015; Sibiko et al. 2018; Vasilaky et al. 2020). There has also been some theoretical and empirical research on the "Achilles heel" of IBI systems of basis risk under extreme weather variability, and the resulting high costs (Jensen et al. 2016, p. 25). On the other hand, numerous studies have highlighted the benefits of IBI to encourage farmers to invest in smart and sustainable agricultural innovations (Adegoke et al. 2017; Hess and Hazell 2020; Hazell et al. 2021). The regulatory incentives to IBI schemes under the existing EU Common Agricultural Policy and the potential of insurance technologies ("insurtech") to overcome the main obstacles of IBI have not been investigated so far, other than on Internet platforms, such as Blockchain Climate Risk Crop Insurance[4], Global Index Insurance Forum[5], Social Fintech[6] and The Digital Insurer[7], with the notable exception of Jha et al. (2021). In this paper, we aim to fill this research gap by highlighting the opportunities of DLT as a prerequisite for lower premiums and lower transaction costs (e.g., time savings in compensation payment) and the possibilities for innovative national risk management schemes under the post-2020 CAP, which are mentioned as action needed to "support for insurance schemes" and to accelerate "innovation" by the policy brief of the ongoing EU project "SURE Farm"[8].

## 3. Data Description

For the purposes of this research, the authors gathered a set of data on the following issues: economic damages from weather and climate-related extreme events for the period 1997–2017 in the EU-28 countries; insurance and compensation systems in the EU and Switzerland as of 2019; DLT-related cost and time savings for insurance services.

The first part of this data was retrieved from the European Environmental Agency (EEA) and is based on the methodology the latter uses to establish damages with regard to geothermal (e.g., earthquakes, tsunamis and volcanic eruptions), meteorological (e.g., storms), hydrological (e.g., floods and mass movements) and climatological events (e.g., heatwaves, cold waves, droughts and forest fires). The second part of the data (insurance and compensation systems in the EU and Switzerland) was gleaned from existing scientific research and publications (e.g., Palka and Hanger-Kopp 2019, p. 2; and Vroege et al. 2019, p. 105). Finally, the data relating to the savings associated with applications of DLT-based solutions were compiled from reports prepared by various consulting agencies and research institutions (e.g., PwC 2016; KPMG 2017).

**4. Climate Change—A "Window of Opportunity" for the Insurance Sector?**

Climate change is associated with adverse impacts that have already made their presence amply felt: increased temperatures, melting ice caps and rising sea levels. Against this background, the international community (the United Nations, UN, in particular) is paying due attention to this problem and taking steps toward establishing a common legal framework with incentives to combat climate change and adapt to its consequences (UNFCCC 2015). In 2018, the global economy faced losses of 225 billion USD resulting from natural disasters and extreme weather events. This level is ten times higher than in 2000, and the year 2018 itself was the third year in a row with actual losses in excess of 200 billion USD. It is important to note that only 40% of these losses were covered and compensated for by insurers (Aon Benfield 2019).

The world community is currently on track for a global temperature rise of 2.7 °C elsius by the end of the century (UNEP 2021). Hence, it is crucial not only to ensure adequate adaptation to climate change but also to reduce people's exposure to natural hazards and extreme weather events; this requires appropriate measures and sufficient financial resources. According to estimates from the UN, global annual expenditure needs for adaptation to climate change ranges between 140 billion USD and 300 billion USD. By 2050, the cost of adaptation to climate change could reach between 280–500 billion USD. In fact, however, only around 22 billion USD are currently being collected annually for the purpose of adaptation to climate change (Micale et al. 2018). At the same time, climate-related disasters are linked to almost 100 billion USD in annual losses. Moreover, such events could have increasingly serious social and economic consequences. For example, the number of climate-induced migrants is steadily increasing, and considering the current path of global warming, millions of people could be forced to leave their homes and regions in the coming decades due to adverse environmental conditions (IOM 2009). In view of this, another important agreement was signed in 2015 under the auspices of the UN to reduce the risks of climate-related disasters: the Sendai Framework on Disaster Risk Reduction (SFDRR). This framework covers the period between 2015 and 2030 and is aimed at protecting both people's lives and critical infrastructure (the energy sector, transport, agriculture, etc.) (UNISDR 2015).

According to the methodology used by the European Environmental Agency (EEA 2021), there are three major groups of weather and climate-related extreme events that might cause economic damages: meteorological (e.g., storms), hydrological (e.g., floods and mass movements) and climatological events (e.g., heatwaves, cold waves, droughts and forest fires) (see Figure 1). In the period 1980–2019, weather- and climate-related extreme events were responsible for about 81% of total economic losses caused by natural hazards in the EEA member countries, amounting to 446 billion EUR or the equivalent of 11.1 billion EUR per year. Cumulative deflated losses correspond to almost 3% of the GDP of the countries analyzed. However, since a relatively small number (3%) of individual events were responsible for a large share (>60%) of economic losses, there is a high variability of losses from year to year, which makes it difficult to identify a trend. A decade-by-decade view enables a process of smoothing that reveals trends. Average annual (inflation-adjusted) losses were about 6.6 billion EUR in 1980–1989, 12.3 billion EUR in 1990–1999, 13.2 billion EUR in 2000–2009 and 12.5 billion EUR in 2010–2019.

Hence, climate change is already responsible for an increasing amounts of material losses to the economy, the financial markets and society as a whole, most of which are uninsured, even in a prosperous continent such as Europe. In fact, according to the data provided by NatCatSERVICE, Eurostat and MunichRe, coverage for climate-related losses is insufficient, the best results having been achieved by the United Kingdom (UK), where insured losses accounted for over 70% of the total. The most critical situation with regard to covering climate-related risks and losses was identified in Greece, Portugal, Poland and Italy, where damages from climate-related events remained almost uncovered. At the same time, very good rates were achieved by Belgium, Denmark, Lichtenstein and Luxembourg—where over 58% of the losses were insured. Germany, France, Ireland,

Iceland and Switzerland were able to cover almost 50% of the damages caused by climate-related extreme events and natural disasters.

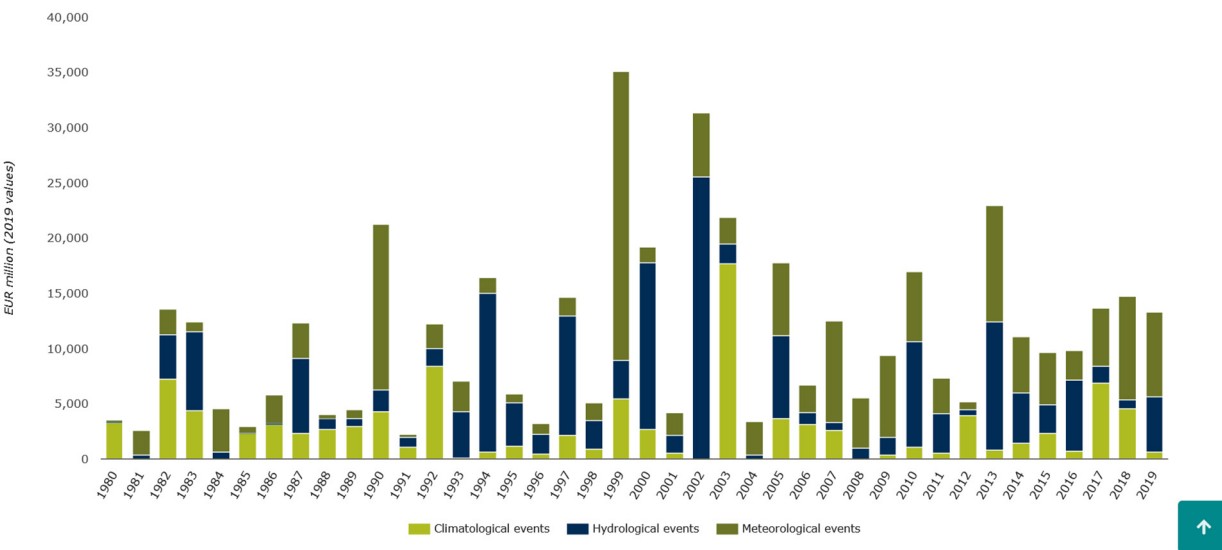

**Figure 1.** Economic damages caused by weather and climate-related extreme events in Europe (EU-28, 1980–2019). Source: EEA (2021). Reprinted with the permission from EEA. The standard EEA's copyright legal notice applies.

The Intergovernmental Panel on Climate Change (IPCC 2022) predicts that climate-related extreme events will become even more frequent around the world. This could indirectly affect multiple sectors and cause systemic failures across Europe, leading to even greater economic losses.

As a consequence, more effective measures are required to prevent global climate-related risks from climate change, to enhance climate adaptation in developing countries and to improve resilience in infrastructure (especially critical infrastructure) in Europe. Additionally, there is an urgent need for innovative financial products and instruments to support the above-mentioned measures, the overarching aim being to provide access to the market of private climate finance (EEA 2021).

A lack of climate-related data and difficulties related to the evaluation of non-financial risks are the main obstacles blocking the way to extending insurance services and providing access to them. Various initiatives are seeking to resolve this issue by formulating recommendations on how to improve data collection, processing and reporting related risks. For instance, in 2021 a special "Task Force on Nature-related Financial Disclosures" was created to support businesses in evaluating emerging nature-related risks and help achieve nature-positive outcomes (UNEP-FI 2021).

## 5. Agricultural Insurance—Status Quo and Prospects for the European Union

The European Union (EU) insurance landscape for agriculture is diverse, as member states are exposed to different types of risks, and their political and social settings differ as well. As a consequence, the "risk management toolbox" of the Common Agricultural Policy (CAP) allows for the use and public support of an array of instruments, including insurance, mutual funds and disaster relief programs. Public support is capped to 70 percent of premiums or annual payments into mutual funds (European Commission 2017). Such public–private partnerships are being promoted to avoid the financing of damage and mitigation measures becoming the sole responsibility of the public sector (Miranda et al. 2018). Although the concept of public–private partnerships for agricultural risk is acknowledged in principle in the management tools of the CAP, member states can exercise wide discretion in their use. Three-quarters of EU countries, including France, Italy, Spain, Austria and the Netherlands, provide subsidies of between 45 and 65 percent for insurers'

so-called multi-risk policies, which cover weather-related risks, including droughts (see Table 1).

**Table 1.** Insurance and compensation systems in the EU-27 and Switzerland.

|  | Hail | Storm | Heavy Rainfall | Frost | Drought |
|---|---|---|---|---|---|
| Belgium [1][3] | X | X | X |  |  |
| Denmark | X | X | X |  |  |
| Germany [2] | X | X | X | X | X |
| Italy [1][3] | X | X | X | X | X |
| Croatia [1][3] | X | X | X | X |  |
| Luxembourg [1][3] | X | X | X | X | X |
| Latvia [1][3] | X | X | X | X |  |
| Lithuania [1][3] | X | X | X | X | X |
| Netherlands [1] | X | X | X | X | X |
| Austria [1][2][3] | X | X | X | X |  |
| Poland [1][3] | X | X | X | X |  |
| Spain [1][2][3] | X | X | X | X | X |
| Switzerland [1][2][3] | X | X | X | X [4] | X |

Note: [1] Multi-peril insurance, [2] IBI, [3] state subsidies [45–65%], [4] snow pressure. Source: Authors' compilation based on Grant (2010). Austria, Italy and Switzerland were taken from Palka and Hanger-Kopp (2019, p. 2), ISMEA (2020) and Vroege et al. (2019, p. 105).

Demand is high, especially because risk premium subsidies provided for the insurance premium are disbursed from national and EU funds. Simplified field loss assessment procedures are helping to enhance insurance density. In the Netherlands and Luxembourg, for example, agricultural yield losses in the field are determined by evaluating the dried parts of the plant, the sizes of the cobs (in the case of maize) or the weight of the grains. In the Netherlands, more than a quarter and in Luxembourg almost every second hectare of the affected areas are already insured against drought damage. In Germany, however, a lack of subsidies and an insurance tax of 19% on insurance premiums for droughts make risk protection a thoroughly unattractive option. In almost all other EU countries, the tax rate is zero or near zero. Italy protects its farmers against weather risks with some 1.6 billion EUR, and France with 600 million EUR. Only Germany, Denmark, the Netherlands and a few others leave this risk to their farmers.

In some countries around the world, e.g., the United States, crop insurance is the primary policy mechanism for reducing agricultural yield and/or income risk (Mahul and Stutley 2010). In Europe, the perceived policy importance of agricultural insurance is lower for several reasons. One of these is that risk management is part of the "second pillar" of the CAP, even if it involves much lower funding than the "first pillar" of direct payments.[9] Moreover, the "second pillar" of risk management competes with other rural development objectives, including the goals of "greening" agriculture and achieving balanced territorial development. The allocation of funds for risk management, therefore, varies depending on the perceived political urgency of direct payments to farmers, the perceived urgency of a "green" and "just" transition in rural areas and the (expected) need for relief payments after disasters.

This is reflected in the current CAP reform debate within the EU. Although the CAP (2023–2027) contains objectives that explicitly emphasize the importance of "other forms of risk management" (see Appendix A to this article), their financing is discussed solely in terms of the "first pillar" as member states' activities, whereas the "second pillar" focuses only on reducing greenhouse gas emissions from agriculture and achieving a balanced territorial development ("just transition") under the EU Green Deal. We view this as an expression of a structural deficit in climate adaptation and disaster risk reduction elements in the CAP (cf. SURE Farm 2021) and in the EU Green Deal and the EU's COVID-19 recovery program (Next Generation EU, NGEU) (cf. Michalek et al. 2020; Schwarze and Sushchenko 2021). For the time being, then, progress on the dissemination

of risk management instruments in EU agriculture can only be expected from initiatives undertaken by member states, such as the recent push from France to renew its 600 million EUR mutual fund (cf. Pistorius 2021).

## 6. Yield-Based vs. Index-Based Insurance

As a rule, climate catastrophes arrive unexpectedly, and the damage caused by such events is not predictable in any precise way. The "classical" insurance techniques and instruments are often not sufficiently effective to solve this problem, as the contractual mechanism for compensation works on the basis of yield losses that have been observed in the past. In practice, the main problem in claims management is that it often takes months to determine and settle payments—months during which losses can rise further. For instance, long-lasting high moisture content in crops could affect infrastructure conditions, e.g., by reducing their drying capacity and making them vulnerable to possible subsequent frost damage. Existing yield-based approaches to insurance for climate-related risks in agriculture have two main drawbacks: fraud detection and risk modelling. Agricultural businesses and farmers tend to overestimate their real losses and claim higher compensation from the insurance companies. Hence, claims management becomes very difficult and requires additional expenditure (both in terms of cost and time) to determine and verify an appropriate amount of compensation for the clients. The second negative feature of a yield-based insurance relates to the modelling of risks, especially given that the average surface temperature on Earth is rising faster than expected, making forecasting unprecedentedly difficult.

Nowadays, ex post and ad hoc compensation is becoming more and more expensive: during 2014–2020, more than 65% of insurance premiums were paid out by the EU as part of the Common Agricultural Policy (Hochrainer-Stigler and Hanger-Kopp 2017). In addition, yield-based insurance may not even be applicable in certain areas: grasslands, for example, are subject to a different number of harvests each year and a very small difference in damages depending on the seasonal frequency of extreme weather events. In such cases, IBI could be considered the most appropriate solution (Hochrainer-Stigler and Hanger-Kopp 2017). IBI relies on the application of physical indicators (temperature, soil moisture, etc.) as a "trigger" for compensation. Compared to yield-based insurance, IBI has several positive features (see Table 2).

**Table 2.** Yield-based vs. index-based insurance.

| Insurance Type | Strengths | | Weaknesses | |
|---|---|---|---|---|
| **Yield-based** | - | long-established, time-honoured | - | high transaction costs |
| | - | holds the "promise" of full risk transfer | - | fraud detection |
| | - | more objective, better reflects climate change | - | high subsidies needed for affordability |
| **Index-based** | - | lower costs, less time needed for compensation (no field loss assessments) | - | lack of reliable data |
| | | | - | advanced technological and modelling capacities needed |
| | - | eliminates bureaucracy | - | high basis risk |
| | - | improves transparency | - | need to bundle small risks |

Source: Authors' own assessment.

First, the IBI approach is more objective and better adapted to climate risks because indicators depend only on the physical properties of the environment. In addition, compensation is limited to a predetermined amount of money calculated on the basis of previous events and their associated losses. Another significant advantage of IBI is improved trust between insurance companies and their clients. At the same time, IBI could simplify field loss assessment, reduce bureaucracy and increase transparency—making it less costly for customers such as small-scale farmers (Gommes and Kayitakire 2013). Despite all its

positive features, the implementation of IBI is also associated with certain obstacles: a lack of reliable data, technical and modelling requirements and the large remaining basis risk of up to 70% in case of severe weather events (cf. Jensen et al. 2016). Changing risk patterns in the case of abrupt climate change could also jeopardize the application of IBI. In addition, the premiums per farmer are small so that insurance companies usually have to aggregate risks in order to transfer them to the reinsurer (Hess and Syroka 2005). The yield-based approach is a long-established practice of agricultural insurance. Although it holds the promise of full compensation, in practice it requires some retention of basis risk by those insured in order to avoid the transaction costs of fraud control; due to the higher costs involved, it comes with higher subsidies to make it affordable for customers.

In contrast to yield-based insurance, in which indemnities are paid based on evidence or crop loss estimates, policyholders in an index insurance policy are compensated on the basis of the realization of an objective index, e.g., precipitation, accumulated temperature or regional average crop yield, which is closely correlated with yield losses at the farm level (Bielza et al. 2009, p. 45). As the indices are measured by government agencies or other third parties and are thus objective and independent, this insurance is well-suited for farmers with a highly index-correlated risk structure (Bielza et al. 2009, p. 33). Index-based insurance payouts are linked to yield, satellite or weather triggers, such as rainfall, temperature, humidity and crop yields, which serve as proxies for losses or similar data sources (Weingärtner et al. 2017, p. 14; Gommes and Kayitakire 2013). Compensation is paid as soon as one of these triggers deviates from a predetermined value. In other words, products are insured against events that cause losses and not against actual losses in the fields (Hochrainer-Stigler and Hanger-Kopp 2017, p. 4).

The selection of the index on which the insurance should be based has been widely discussed (Maestro Villarroya 2016, p. 12). Two types of index insurance are mentioned in most of the scientific literature. The first is area yield index insurance (AYII) and the second is weather-based index insurance (WII) (cf. World Bank 2011, p. 9ff.). There are different types of drought indicators that can be used as indices in drought insurance. These include the meteorological, agricultural, hydrological and socioeconomic drought indicators mentioned above. Index-based insurance is considered by some to be the best solution for developing countries (Rao 2010, p. 193).

Careful selection and the use of a combination of indices are crucial in order to identify production risks in agriculture. In the agricultural sector, AYII and WII are among the most common index-based insurances (Maestro Villarroya 2016, p. 12). In addition, these can be supplemented by remote-sensing-based index insurance (Coleman et al. 2018). The following types of indices exist for crop insurance: area yield trigger, meteorological trigger, the vegetation index and combinations of different factors (as described below).

## 7. Index-Based Area Yield Insurance

Under index-based acreage yield insurance (AYII), compensation is paid to all insured farmers in a defined area (e.g., a county) when zonal yields which have been achieved in the past in a homogeneous geographic area fall below a certain threshold value. The yield insured is determined as a percentage (usually 50 to 90 percent) of the average yield for the area. Compensation is paid regardless of the actual losses suffered if the average yield on the farmer's land is less than the average yield on the area. AYII is similar to MPCI but refers to the average yield per hectare (World Bank 2011, p. 9; Mahul and Stutley 2010, p. 75; Maestro Villarroya 2016, p. 12). For the successful use of AYII, the yield of a farmer should correlate strongly with yield per unit area (Bokusheva and Breustedt 2012, p. 136). Due to the fact that the exact causes of loss cannot be identified, it is difficult to rule out hazards. As a consequence, the AYII provides reasonable protection for farmers, as it covers a wider range of hazards (World Bank 2011, p. 11). The AYII requires historical yield data, which can be used to determine the average normal yield and the insured normal rate of return (Mahul and Stutley 2010, p. 75). In addition, current yield data are necessary to assess the level of compensation when yield losses occur (Coleman et al. 2018, p. 11). This type

of index insurance is used in the U.S., Sweden, Canada, India, Mexico and several other countries (Maestro Villarroya 2016, p. 12; Mahul and Stutley 2010, p. 78).

## 8. Weather-Based Index Insurance

Indemnities are paid under weather-based index insurance (WII) as soon as a specific type of weather or hydrological threshold is met (Maestro Villarroya 2016, p. 12). These thresholds must be met over a specified period of time and evidenced by the data measured at a weather station. Once the realized value of the index exceeds or falls below a predefined threshold, a compensation payment is triggered. This may occur, for example, in the event of too much or too little precipitation, which is expected to lead to crop failure. This compensation is calculated on the basis of a pre-agreed sum per index unit (for example, dollars per millimeter of rainfall) (World Bank 2011; Mahul and Stutley 2010, p. 75). As with AYII, a high correlation of returns with the weather-based index is an important prerequisite for the applicability of WII. This indicates the high potential for risk mitigation with the respective weather-based insurance scheme (Bokusheva and Breustedt 2012, p. 136). The hazards mainly covered by WII include precipitation shortfalls and surpluses due to high, low or prolonged temperatures. Thus, it covers meteorological, agricultural and hydrological droughts. Other hazards can include strong winds and sun, and a combination of these (World Bank 2011, p. 10). For the WII, as for the AYII, ground data are needed to establish the index and to draw up the contract. The WII, which is based on ground measurements, relies on current weather data and historical data, in addition to some agricultural data, to design and calibrate products (World Bank 2011).

## 9. Remote-Sensing-Based and Combined Drought Indicator Index Insurance

Another option for hedging against drought risks is satellite-based index insurance or remote-sensing-based index insurance. Peled et al. (2010), Sepulcre-Canto et al. (2012), Gommes and Kayitakire (2013), among others, proposed the use of remote-sensing-based index insurance to complement AYII and WII. Due to the sparse network of meteorological weather stations in developing countries and some southern European countries (e.g., in the Western Balkan region), there is often a lack of precipitation data. Satellites can be used to fill this gap and to provide timely information on actual crop development for the structuring of the index insurance (Gommes and Kayitakire 2013, p. 238). In the case of AYII, according to Gommes and Kayitakire (2013, p. 52f.), it should be possible to determine the regional yields of homogeneous cropping patterns for various crops with the help of remote sensing technologies. Likewise, remote sensing data on vegetation status can be used as additional input parameters for plant growth models and may thus also be useful for yield assessment (Gommes and Kayitakire 2013, p. 52f.).

Both the WII and AYII rely largely on ground-based measurements (Coleman et al. 2018, p. 7). In the case of remote-sensing-based index insurance, satellite data are used to supplement ground-based data indices or to develop remote sensing index insurance products. Instead of measuring directly on the ground, various data, such as precipitation estimates, evapotranspiration, vegetation indices and soil moisture, which are based on specific biophysical dynamics, are collected using satellites. In order to obtain the datasets, the satellites are used to measure the required data and are calibrated with some additional soil information. The index is designed in such a way that it makes yield loss practical based on the 29 parameters used (Coleman et al. 2018, pp. 14, 30; Gommes and Kayitakire 2013, p. 52). In the academic literature, the normalized difference vegetation index (NDVI) is most commonly used in remote-sensing-based index insurance. This index assesses the spatial and temporal variability of vegetation and generates indices using time series reconnaissance imagery. It is based on the principle of detecting changes in vegetation caused by temporal variations in soil moisture which are then captured using NDVI (Peled et al. 2010, p. 271). NDVI measures the ability of red and near-infrared light (NIR) to be differentially absorbed by leaves and correlates linearly with the amount of synthetic active radiation absorbed by plants (Peled et al. 2010, p. 271). This technology can be used in

many parts of the world (World Bank 2011, p. 18). When it comes to hedging against drought risks, satellite-based index insurance is a very cost-effective method, especially for rainfed crops. These include, for example, pastures or large monocultures. According to Maestro Villarroya (2016, p. 17), the technological advances in satellite remote sensing enable accurate measurements to be made at specific spatial scales and spectral bandwidths that allow for dynamic monitoring of environmental conditions such as vegetation cover. Due to the more precise measurements, remote-sensing-based index insurance can thus be used as an efficient tool for evaluating growing conditions and crop drought (Maestro Villarroya 2016, p. 17).

The improvements in precise measurement achieved by remote-sensing-based index insurance have been confirmed by Peled et al. (2010). In their empirical study, Peled et al. (2010, p. 271) compared the performance of soil drought indices in Europe using remote sensing of vegetation. They were able to find the most suitable drought index based on the highest correlation with NDVI. Their work complements numerous attempts made in recent years to create global datasets of soil moisture and drought indices (Peled et al. 2010, p. 271). Using these tools, the interannual variations in drought indices were compared with the interannual changes in vegetation, which in turn were captured by the NDVI (Peled et al. 2010, p. 271). Subsequently, the correlations of five drought indices (Palmer drought index (PDSI), self-calibrating PDSI (SC-PDSI), standardized precipitation index (SPI) and normalized total surface soil moisture depth (NSMS)) with the NDVI were assessed. In comparing the five drought indices, Peled et al. (2010, p. 271) noted that, due to its optimal correlation with the NDVI, the NSMS index is best suited to describe the actual changes in vegetation most realistically. Especially in areas where hot and dry summers occur, the correlation between annual variations in NDVI and drought indices is highest (Peled et al. 2010, p. 276). Peled et al. (2010) concluded additionally that reliable data for soil moisture can be collected through advances in satellite technology, as they provide long-term weather forecasts and thus enable more efficient preparations for drought. Similarly, Swiss Re (Andriesse 2019) has developed an innovative solution using a soil moisture deficit index developed for drought-related losses. In this approach, satellites are used to measure soil moisture so that, thanks to full digitization, the insurance process can subsequently be carried out using blockchain technology. This can be used for rainfed crops worldwide and is already being deployed in some European countries. NDVI-based insurance programs are currently being used in Canada as pilot projects in pasture production, in Spain for drought in pastures of livestock breeders and in the U.S. (Maestro Villarroya 2016, p. 17). In Kenya, the government has implemented the first index-based livestock insurance intervention as a component of the Kenya Livestock Insurance Program (KLIP). Andrew Mude, the inventor of this tool, received the 2016 World Food Prize as an acknowledgement of his efforts (Russell 2020).

Summing up, from the EU's perspective, IBI could bring more benefits than drawbacks. However, there is no market for related schemes across Europe, and risk management is not unified across the EU (Ramsey and Santaremo 2017). In other words, on the path toward an EU-wide application of IBI, two problems should be kept in mind: the cost of implementation could be enormous, and basis risk could exacerbate the problems of market acceptance (IFAD 2017).

## 10. DLT for a Better Agribusiness and Related Insurance Products

In recent decades, precision technologies and smart contracts have entered the agrifood systems (AFS) of this world (Xu et al. 2020; Stranieri et al. 2021). The advent of modern agricultural technology such as sensors, the Internet of Things, enabled smart devices and smart contracts provides a promising foundation for what might be termed "agriculture 4.0" and for the creation of "smart AFS." The aim of smart AFS is to improve the efficiency of the food chain in relation to physical (e.g., climate and soil), technical (sensors and machines) and business (sales contracts, insurance) factors. The best (i.e., most efficient) response of smart machines to, for example, climate extremes (e.g., water

scarcity) depends on communication among enabled smart devices with other intelligent nodes of the production, sales and risk management systems in the network of agriculture and food production. Smart machines collect information about an unfolding climatic event, and broadcast it to other machines in the field and to nodes along the supply chain. The goal of the Internet of Agri–Food (IoAF) is to send out messages about system-threatening events—such as soil moisture extremes—to active cropping technologies, to those generating crop loss assessments and environmental hazards reports, to cooperative financial risk management, sales and storage agencies, to neighboring farmers and to insurance companies—in little time with high accuracy, in other words: lowering the transaction cost.

Nowadays, a huge amount of data must be processed to cover the needs of insurers (and those insured), at least in the two above-mentioned areas. Moreover, in the modern world, data protection is becoming an increasingly important issue for all economic operators. For this reason, companies and governments from different countries are looking more closely at the opportunities offered by DLT. A starting point has been elaborated based on blockchain (currently the most popular type of solution). Despite the fact that this technology has some limitations (e.g., the number of operations it can handle within a given period of time), the level of data protection is high enough to reduce significantly the risks of external interventions to obtain data or important business information (e.g., "hacking"). Additionally, a combination of DLT with artificial intelligence (AI), the Internet of Things (IoT), big data and other innovations could offer unprecedented breakthroughs for the entire insurance sector. For instance, DLT is crucial to ensuring data integrity and securing user authentication and authorization in a trusted IoT system (Kumar and Sharma 2021). Moreover, a combination of IoT with DLT is crucial to providing reliable data on supply chain tracking in agriculture (Ronaghi 2021). That is why "insurtech" is not just a modern trend but has already become an important part of the daily business activities of various economic sectors (see Table 3).

Several important benefits have been identified for a DLT-based application of IBI, such as improved real-time exposure assessments and enhanced accident and risk prediction. These benefits contribute toward improving data processing and also facilitate understanding of the scenario-based assessments of different changing parameters in real-time contexts.

DLT could bring significant cost and time savings, including reduced transaction costs (e.g., time for negotiations and quotations). According to available estimates, an implementation of DLT solutions for the insurance sector could reduce the time required for negotiations and quotations by up to 90% (Generali 2018). As a result, reinsurers could make the process of reserve estimations easier and establish so-called "streamlined reinsurance" operations. However, the most important advantage for all insurers is improved liquidity control.

Insurtech facilitates deeper risk assessments, offers more sophisticated preventive models, improves interactions, enhances operational capabilities and makes efficient use of ecosystem and market resources (i.e., lower transaction costs). According to the findings provided by PwC (2016), the most important opportunity for insurers arises from self-directed services (e.g., customer acquisition and customer services) and usage-based insurance (e.g., pay-as-you-go).

Moreover, a variety of operational benefits for agricultural insurance relate to improved claims management: a coordinated and synchronized overview and verification of transactions and other information; enhanced third-party transactions (e.g., "claim leakage"); enforced fraud detection and better alignment with new legal requirements for financial institutions. Such improvements could create additional benefits through behavior-based underwriting (e.g., pay-as-you-go). Additionally, existing enhanced requirements for the financial market (e.g., Basel III, Directives 2016/2341, 2017/828) impose certain limitations on the activities of financial institutions. Here, it is not only insurance companies that should comply with the requirements when providing their services, but

other institutional investors as well. The new legal requirements on the financial market are indeed forcing institutional investors to analyze and evaluate non-financial risks when making their investment decisions.

**Table 3.** DLT-related cost and time savings for insurance services.

| | Area of Application | Practical Cases | Time/Money Savings |
|---|---|---|---|
| **Signing the contract and execution** | Smart contracts | R3, CatBonds, CatSwaps, Sprout Insure[10], Etherisc (Kim and Laskowski 2017, p. 13) | up to 2–3 days, no escrow cost[11], reducing the costs of issuing contracts by 41% |
| **Microfinancing** | Peer-to-peer insurance | Lydia, Everex[12] | average cashback of 30% of the premiums[13] |
| **Claim management** | Fraud detection | Shift Technology (Claims automation) | "hit-rate" more than 2.5 times better than standards[14], reduction of annual losses of up to 10% decrease in claim cycles from three months to one week (Sprout Insure) |
| **Underwriting** | Behaviour-based underwriting | Atidot | identification of up to 25% under-insured policies[15] |
| **Parametric insurance** | Mechanism selection | Kenyan Livestock Insurance Program (KLIP) | up to 2–3 months |
| **KYC ("Know your client") and AML (Anti-Money-Laundering Laws)** | Due diligence | InterchainZ | up to 90% of time up to 8 billion USD[16] |
| **Risk transfer** | Reinsurance | B3i (Aegon, Allianz, Munich Re, Swiss Re and Zurich Re) | 15–20% expenses[17] |

Source: Authors' own compilation.

Additionally, a set of market benefits associated with the application of DLT in the insurance sector reflects new business opportunities. The most important improvement could be better access to insurance services for small and medium-sized clients. Here, insurers could drastically reduce their administration costs and make their services more accessible to those who have previously been excluded from classical schemes due to the negative cost–benefit ratios of the insurance products provided. Further, DLT creates a common platform for all the key participants of the insurance process and improves the efficiency of their communications. Such a platform could be considered as a common workspace to enable all involved to track and understand the quotations workflow. DLT applications can trigger innovation in the agricultural ecosystem beyond smart insurance contracts, such as improving food protection, supply chain reliability, the traceability of origin and trade performance, which would benefit both farmers and consumers.

Even despite all the above-mentioned benefits, some difficulties are associated with DLT in relation to IBI-based agriculture insurance products. First, privacy challenges exist for data analysis, since there are very complex and challenging data protection laws in force in the EU. The second important obstacle to applying DLT in agricultural insurance services is associated with the different regulations that exist within different jurisdictions: this could pose some obstacles because different legal acts apply simultaneously to one and the same chain operation. Another challenge is associated with the decentralized storing of data—no one person or entity is responsible for the data chain stored.

## 11. Conclusions

The application of yield-based insurance schemes in agriculture has proven to be less effective than index-based solutions. This disadvantage is related primarily to the time gaps that exist between an actual event and the payment of compensation. Additionally, it is very hard to estimate actual losses. Lack of trust between economic operators could be regarded as one of the reasons for this. Moreover, yield-based insurance products are



relatively expensive and not accessible to small customers. From the point of view of the underwriting process, there are a number of options for replacing yield-based insurance with IBI—solving the above-mentioned problems by introducing an independent and objective physical "trigger" to facilitate quick compensation payments to clients is key to making these options workable.

DLT solutions on the crypto-currency market demonstrate some positive features of this technology and offer prospects for its application in other segments of the financial market. When using insurtech with index-based insurance in agriculture, it is important to consider some of its specific aspects. For instance, DLT could offer an improved real-time exposure assessment, facilitate accident and/or risk forecasting and assist with reserve calculations for reinsurance. Furthermore, this technology could be used when implementing behavioral underwriting.

**Author Contributions:** Conceptualization, R.S. and O.S.; methodology, O.S.; investigation, R.S. and O.S.; resources, R.S.; data curation, O.S.; writing—original draft preparation, O.S.; writing—review and editing, R.S.; visualization, O.S.; supervision, R.S.; project administration, O.S.; funding acquisition, R.S.. All authors have read and agreed to the published version of the manuscript.

**Funding:** This research received no external funding.

**Data Availability Statement:** The Figure 1 supporting dataset is publicly archived at https://www.eea.europa.eu/ims/economic-losses-from-climate-related (accessed on 8 March 2022).

**Conflicts of Interest:** The authors declare no conflict of interest.

## Appendix A. Risk Management Tools of the CAP Post-2020 Reform

### Article 70

1. *Member States* shall grant support for risk management tools under the conditions set out in this Article and as further specified in their CAP Strategic Plans.
2. *Member States* shall grant support under this type of interventions in order to promote risk management tools, which help genuine farmers manage production and income risks related to their agricultural activity which are outside their control and which contribute to achieving the specific objectives set out in Article 6.
3. *Member States* may grant in particular the following support: (a) financial contributions to premiums for insurance schemes; (b) financial contributions to mutual funds, including the administrative cost of setting up;
4. *Member States* shall establish the following eligibility conditions:

    (a) the types and coverage of eligible insurance schemes and mutual funds;
    (b) the methodology for the calculation of losses and triggering factors for compensation;
    (c) the rules for the constitution and management of the mutual funds.

5. *Member States* shall ensure that support is granted only for covering losses of at least 20% of the average annual production or income of the farmer in the preceding three-year period or a three-year average based on the preceding five-year period excluding the highest and lowest entry.
6. *Member States* shall limit the support to the maximum rate of 70% of the eligible costs.
7. *Member States* shall ensure that overcompensation as a result of the combination of the interventions under this Article with other public or private risk management schemes is avoided.

**Taken from**: European European Commission (2018): *CAP Strategic Plans*. Emphasis by the authors.

## Notes

1.    Index-based insurance is a relatively new but innovative approach to insurance provision that pays out benefits on the basis of a predetermined index (e.g., rainfall level) for loss of assets and investments—primarily working capital—resulting from weather and catastrophic events (IFC 2020).

2.    Yield-based insurance provides compensation equivalent to the difference between the obtained yield and the yield guaranteed at the pre-defined rate at the beginning of the contract (Atlas Magazine 2017).

3.    Financial technology (fintech) is a technological innovation that aims to compete with traditional financial methods in the delivery of financial services. It is an emerging industry that uses technology (e.g., Distributed Ledger Technologies) to improve activities in finance services. A subset of fintech companies that focus on the insurance industry is collectively known as insurtech.

4.    https://www.climatefinancelab.org/project/climate-risk-crop-insurance/ (accessed on 8 March 2022).

5.    https://www.indexinsuranceforum.org/blog/blockchain-application-agriculture-insurance (accessed on 8 March 2022).

6.    https://socialfintech.org/blockchain-crop-insurance/ (accessed on 8 March 2022).

7.    https://www.the-digital-insurer.com/dia/aon-oxfam-etherisc-agriculture-insurance-blockchain-makes-first-payouts/ (accessed on 8 March 2022).

8.    https://www.surefarmproject.eu/wordpress/wp-content/uploads/2020/08/D4.6_Policy-Brief-on-the-CAP-post-2020.pdf (accessed on 8 March 2022).

9.    Between 2003 and 2015 the "first pillar" accounted for an average of approximately 70 percent (68–72%) of total CAP spending (Matthews 2016).

10.    Sprout Insure (Ashley King-Bischof & Sandro Stark). Blockchain Climate Risk Crop Insurance—The Global Innovation Lab for Climate Finance (climatefinancelab.org (accessed on 20 February 2022)).

11.    https://hackernoon.com/smart-contracts-a-time-saving-primer-b3060e3e5667 (accessed on 8 March 2022).

12.    https://blog.everex.io/problems-with-microlending-and-how-blockchain-solves-them-1582f98e2a7c (accessed on 8 March 2022).

13.    https://p2pconference.com/speaker/tim-kunde/ (accessed on 6 May 2020).

14.    https://www.digitalinsuranceagenda.com/180/shift-technology-ai-that-understands-insurance-claims/ (accessed on 8 March 2022).

15.    http://www.oxbowpartners.com/pdfs (accessed on 8 March; search: Atidot).

16.    https://www.jdsupra.com/legalnews/using-blockchain-for-kyc-aml-compliance-25325/ (accessed on 8 March 2022).

17.    https://www.disruptordaily.com/blockchain-use-cases-insurance/ (accessed on 8 March 2022).

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
