# Peer review of "Climate Insurance for Agriculture in Europe: On the Merits of Smart Contracts and Distributed Ledger Technologies"

_jrfm, doi:10.3390/jrfm15050211_

Round 1
Reviewer 1 Report
The authors compare features of yield-based insurance and index-based insurance in agriculture, and show advantages of applying distributed ledger technologies in combination with index-based insurance.
The topic is thematic and interesting. The paper with little quantitative contribution should be categorized into insurance economics. I recommend publishing it after some revision according to my comments below:
(1) The paper overall is very well written, but the presentation is sometimes over repetitive. The authors should work on a more stringent version.
(2) It would be nice to have some quantitative results to justify your conclusion that “applying distributed ledger technologies in combination with IBI could not only resolve existing problems but also facilitate the development of innovative risk management tools...”
(3) Some figures such as Figures 1 and 2 are directly copied from other sources. The authors and editors must check if there are any copyright issues. The authors may think to use links to the cited sources rather than such a direct copy.
(4) Minor sloppiness:
- Page 4, bottom: Change "world ." to "world."
- Page 10, top: Change "table 3" to "Table 3"
Author Response
(1) We have shortened the paper (page 2, 4) and erased Figure 2, including the accompanying text on page 10, to make the paper more stringent.
(2) We have extended the list of innovations made possible by the application of DLT in agricultural insurance, including co-benefits to consumers (page 16). These innovations are supported by the post-2020 CAP according to the SURE farm project's policy brief (page 3), but can not be quantified at this early stage.
(3) Fig. 2 was erased (also to arrive at a more stringent paper, see (1) above)
(4) All minor sloppinesses were corrected.
Reviewer 2 Report
This paper deals with the challenges and opportunities that climate changes pose to insurance domain. The research is essentially descriptive and gives a comprehensive idea of this issue. Some interesting considerations are given about the possibility of implementing DLT in this field.
Some suggestions:
- Please, highlight the contribution of your research to the existing literature ;
- The paper does not comply with the manuscript preparation guidelines given by this journal
Author Response
1. We have highlighted the contribution of our research in a newly added "State of the literatur" section (page 3)
2. The paper will comply to the manuscript preparation guidelines of mdpi after edit.